# *Brucella* suppress STING expression via miR-24 to enhance infection

**Mike Khan[1], Jerome S. Harms[2], Yiping Liu[2], Jens Eickhoff [3], Jin Wen Tan [2], Tony Hu [2], Fengwei Cai[2], Erika Guimaraes[4,5], Sergio Costa Oliveira[4], Richard Dahl[6,7], Yong Cheng[7], Delia Gutman [8], Glen N. Barber[8], Gary A. Splitter[9], Judith A. Smith [2,10]***

1 Cellular and Molecular Pathology Training Program, University of Wisconsin-Madison, Madison, Wisconsin, United States of America, 2 Department of Pediatrics, University of Wisconsin-Madison, Madison, Wisconsin, United States of America, 3 Department of Biostatistics and Medical Informatics, University of Wisconsin-Madison, Madison, Wisconsin, United States of America, 4 Departamento de Bioquímica e Imunologia, Instituto de Ciências Biológicas, Universidade Federal de Minas Gerais, Belo Horizonte-Minas Gerais, Brazil, 5 Programa de Pós-Graduação em Genética, Instituto de Ciências Biológicas, Universidade Federal de Minas Gerais, Belo Horizonte, Minas Gerais, Brazil, 6 Department of Microbiology and Immunology, Indiana University School of Medicine, South Bend, Indiana, United States of America, 7 Department of Biological Sciences, University of Notre Dame, Notre Dame, Indiana, United States of America, 8 Department of Cell Biology, University of Miami, Miami, Florida, United States of America, 9 Department of Pathobiological Sciences, University of Wisconsin-Madison, Madison, Wisconsin, United States of America, 10 Department of Medical Microbiology and Immunology, University of Wisconsin-Madison, Madison, Wisconsin, United States of America

* jsmith27@wisc.edu

**Data Availability Statement:** All relevant data are within the manuscript and its Supporting Information files.

**Funding:** M.K. received the F31 AI115931 National Institutes of Health training award. S.C.O. received

## Abstract

Brucellosis, caused by a number of *Brucella* species, remains the most prevalent zoonotic disease worldwide. *Brucella* establish chronic infections within host macrophages despite triggering cytosolic innate immune sensors, including Stimulator of Interferon Genes (STING), which potentially limit infection. In this study, STING was required for control of chronic *Brucella* infection *in vivo*. However, early during infection, *Brucella* down-regulated STING mRNA and protein. Down-regulation occurred post-transcriptionally, required live bacteria, the *Brucella* type IV secretion system, and was independent of host IRE1-RNase activity. STING suppression occurred in *MyD88*-/- macrophages and was not induced by Toll-like receptor agonists or purified *Brucella* lipopolysaccharide (LPS). Rather, *Brucella* induced a STING-targeting microRNA, miR-24-2, in a type IV secretion system-dependent manner. Furthermore, STING downregulation was inhibited by miR-24 anti-miRs and in *Mirn23a* locus-deficient macrophages. Failure to suppress STING expression in *Mirn23a*[-/-] macrophages correlated with diminished *Brucella* replication, and was rescued by exogenous miR-24. *Mirn23a*-/- mice were also more resistant to splenic colonization one week post infection. Anti-miR-24 potently suppressed replication in wild type, but much less in STING[-/-] macrophages, suggesting most of the impact of miR-24 induction on replication occurred via STING suppression. In summary, *Brucella* sabotages cytosolic surveillance by miR-24-dependent suppression of STING expression; post-STING activation "damage control" via targeted STING destruction may enable establishment of chronic infection.

the NIH R01 AI116453 National Institutes of Health award. G.A.S. received the NIH R01 AI073558 National Institutes of Health award. R.D. received NIH R01 DK109051 National Institutes of Health award. J.A.S. was a multi-PI on NIH R01 AI073558. The funders had no role in study design, data collection and analysis, decision to publish, or preparation of the manuscript.

**Competing interests:** The authors have declared that no competing interests exist.

## Author summary

Cytosolic pattern recognition receptors, such as the nucleotide-activated STING molecule, play a critical role in the innate immune system by detecting the presence of intracellular invaders. *Brucella* bacterial species establish chronic infections in macrophages despite initially activating STING. STING participates in the control of *Brucella* infection, as mice or cells lacking STING show a higher burden of *Brucella* infection. However, we have found that early following infection, *Brucella* upregulates a microRNA, miR-24, that targets the STING messenger RNA, resulting in lower STING levels. Dead bacteria or bacteria lacking a functional type IV secretion system were defective at upregulating miR-24 and STING suppression, suggesting an active bacteria-driven process. Failure to upregulate miR-24 and suppress STING greatly compromised the capacity of *Brucella* to replicate inside macrophages and in mice. Thus, although *Brucella* initially activate STING during infection, the ensuing STING downregulation serves as a "damage control" mechanism, enabling intracellular infection. Viruses have long been known to target immune sensors such as STING. Our results indicate that intracellular bacterial pathogens also directly target innate immune receptors to enhance their infectious success.

## Introduction

*Brucella* spp. are Gram-negative, facultative intracellular α-proteobacteria which cause the zoonotic disease brucellosis [1,2]. Human brucellosis is characterized by an acute undulating fever accompanied by flu-like myalgias before developing into a chronic disease, with long-term pathologies such as sacroiliitis, arthritis, liver damage, meningitis, and endocarditis [3]. Brucellosis in animals often causes orchitis and sterility in males and spontaneous abortions in females, leading to profound economic loss worldwide [4]. During chronic infection, *Brucella* live and replicate within macrophages and other phagocytes. This intracellular localization renders the organism refractory to even prolonged multiple antibiotic treatments, and relapses occur in 5–10% of cases [3]. In the U.S., brucellosis has been largely controlled through vaccination of livestock with live attenuated strains, though outbreaks still occur [5–7]. Currently, no safe and effective human vaccine exists. The mechanism(s) involved in supporting the intracellular persistence of *Brucella* remain unclear.

Innate immune responses form the first line of defense against bacterial pathogens. However, *Brucella* express multiple 'atypical' virulence factors, which stymie innate defenses. For example, *Brucella* spp. resist complement activation and express a weakly endotoxic "smooth" lipopolysaccharide that is a poor agonist for the innate immune sensor Toll-like receptor 4 [8]. Despite sequestration in membranous compartments, *Brucella* trigger cytosolic innate immune sensors including various inflammasomes and the Stimulator of Interferon Genes (STING) [9–12]. STING resides in the endoplasmic reticulum membrane and upon activation by bacterial cyclic-di-nucleotides or cyclic GMP-AMP (c-GAMP), STING translocates to perinuclear clusters where it co-localizes with and activates TANK binding kinase I (TBK1), which in turn phosphorylates the IFN-β regulatory transcription factor IRF3 [13,14]. In addition to Type I interferon induction, STING is essential for optimal induction of NF-κB-dependent pro-inflammatory cytokines and other host defense genes, and regulates autophagy [15]. Evidence from the cancer literature also suggests STING critically supports effective CD8+ T cell adaptive immune responses [16]. Previously, we have shown that STING is required for Type I interferon production in response to infection with *Brucella abortus*, and that STING contributes to control of *B. abortus* infection at 72 hours *in vitro* [9,17].

Here, we report that STING is critical for the control of acute and chronic *Brucella* infection *in vivo*. However, early during infection, *Brucella* down-regulate STING (T*mem173*) mRNA expression and protein. Concurrently with STING suppression, *Brucella* induce a STING-targeting microRNA miR-24. Inhibition by anti-miR-24 or genetic deficiency of miR-24-2 leads to a significant increase in STING expression as well as augmented IFN-β production in macrophages. Inability to induce miR-24 and downregulate STING compromised *Brucella* survival in macrophages and in mice. These results suggest that *Brucella* mitigates the cost of innate immune activation by miR-24-dependent targeting of STING expression.

## Results

### STING is required for chronic control of *Brucella in vivo*

In previous studies, we showed that STING is required for control of *Brucella* replication *in vitro* from 24–72 hours [17,18]. We confirmed that by 24h, STING (*Tmem173*)[-/-] macrophages displayed significantly increased *Brucella* infection (**Fig 1A**). Recently, we had also shown that STING is required for control of *Brucella* infection in mice at 1 and 3 weeks [18]. To confirm these results and evaluate the role of STING in longer-term chronic *Brucella* infection [19], wild type C57BL/6 and STING[-/-] mice were infected with wild-type *S2308 Brucella abortus* for 1, 3 and 6 weeks. Splenic colony forming units (CFU) showed an approximately two-log difference between STING[-/-] mice and age-matched control C57BL/6 mice at 3 weeks and ~1.5 log difference at 6 weeks (**Fig 1B**). These data indicate that STING critically participates in the control of chronic *Brucella* infection *in vivo*.

### *Brucella* infection suppresses STING expression independently of IRE1 endonuclease activity and requires live bacteria and Type IV secretion

Given the requirement for STING in the control of chronic infection, it was surprising to note significant STING (*Tmem173*) mRNA down-regulation in bone marrow-derived macrophages (BMDMs) infected with wild type *16M Brucella melitensis* at 24h (published RNAseq data set in [17]). To confirm the RNAseq data, and determine whether other *Brucella* species down-regulate STING, *v-raf/v-myc* immortalized murine bone marrow-derived macrophages [20] were uninfected or infected with different *Brucella* species for 24 hours and STING (*Tmem173)* mRNA levels assessed via RT-qPCR (**Fig 2A**). *B. melitensis*, *B. abortus* and *B. suis* are human pathogens and primarily infect ruminants, cattle and swine, respectively. *B. neotomae* has been

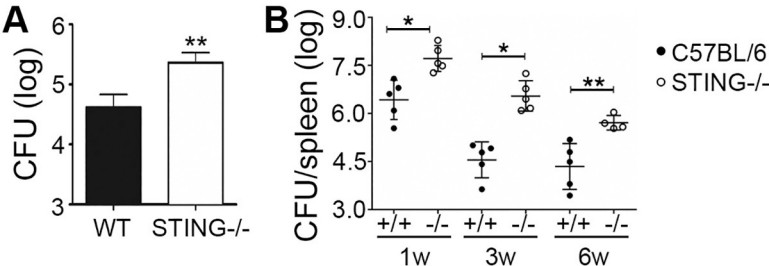

**Fig 1. STING is required for control of acute *Brucella* infection *in vitro*, and acute and chronic infection *in vivo*.** A) Bone marrow derived macrophages from wild type C57BL/6 control (WT) or STING[-/-] mice were infected with 10 multiplicity of infection (MOI) *B. abortus* for 24h prior to enumeration of colony forming units (CFU). Error bars denote triplicate determinations. B) Wild-type C57BL/6 (black circles, +/+) and STING[-/-] mice (open circles, -/-) were infected for 1, 3 or 6 weeks with 10[6] CFU *Brucella abortus 2308* and splenocyte CFUs determined. Circles represent individual mice with 5 mice per group except the STING[-/-] from week 6 (4 mice). Bars denote median CFU/group. Results in (A) and (B) are representative of 3 independent experiments.

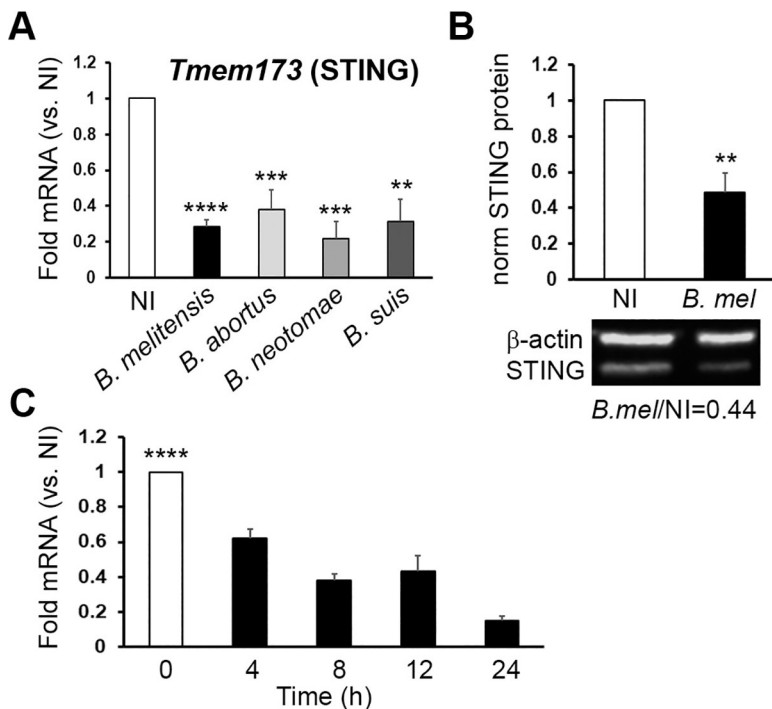

**Fig 2. *Brucella* suppresses STING expression.** A) Immortalized murine bone marrow derived macrophages were not infected (NI) or infected with *B. melitensis* 16M (black bars), *B. abortus* (light gray), *B. neotomae* (medium gray) or *B. suis* (dark gray) as indicated at 100 MOI for 24h. Cells were lysed, RNA isolated and reverse transcribed, and relative *Tmem173* (STING) mRNA levels determined by quantitative PCR (qPCR) with normalization to 18S rRNA and uninfected controls (NI, set = 1). Results are from 25, 7, 5, and 5 independent experiments respectively, with error bars denoting SEM. P-values are vs. NI control. B) Protein expression of STING: cells were infected with 100 MOI of *B. melitensis* for 24h, and lysates resolved using SDS PAGE. STING and β-actin proteins were detected by western blot. Band fluorescence was quantitated and results are means +/- SEM of 5 independent experiments. An example western blot is below the graph with the ratio of β-actin normalized STING fluorescence for *B. melitensis* vs. NI. C) Time course: Cells were infected with 100 MOI *B. melitensis* for the times indicated and processed for RNA quantitation as in (A). Results are means +/- SEM of 9 independent experiments and for all times tested, $p<0.001$ for NI vs. infected samples.

isolated from wood rats and voles but also has been isolated in human neurobrucellosis [21]. The four species of *Brucella* significantly down-regulated STING mRNA compared to uninfected macrophages. STING protein levels also decreased in cells infected for 24 hours with *Brucella* (**Fig 2B**). In macrophages infected with *B. melitensis*, *Tmem173* mRNA down-regulation was evident by 4h post-infection (**Fig 2C**).

*Tmem173* down-regulation required live bacteria, consistent with an active bacteria-driven process (**Fig 3A**). *B. melitensis* with mutations in the type IV secretion system (deletion of the critical VirB2 subunit [22]) displayed an intermediate phenotype with only modest downregulation of *Tmem173*, suggesting an intact type IV secretion system (T4SS) is required for full STING suppression (**Fig 3B**). Regarding the mechanism of suppression, one straightforward possibility was that *Brucella* infection suppresses the activity of transcription factors required for *Tmem173* promoter activity. To address this possibility, we utilized a murine STING-promoter driven luciferase reporter (**Fig 3C**). *Brucella* infection increased the activity of the STING promoter-driven construct (at least 1.5-fold in 3 experiments and a 4-fold increase in one experiment). Heat-killed *Brucella* treatment also increased promoter-driven luciferase activity. This result suggested *Brucella* suppressed *Tmem173* expression downstream of promoter activation.

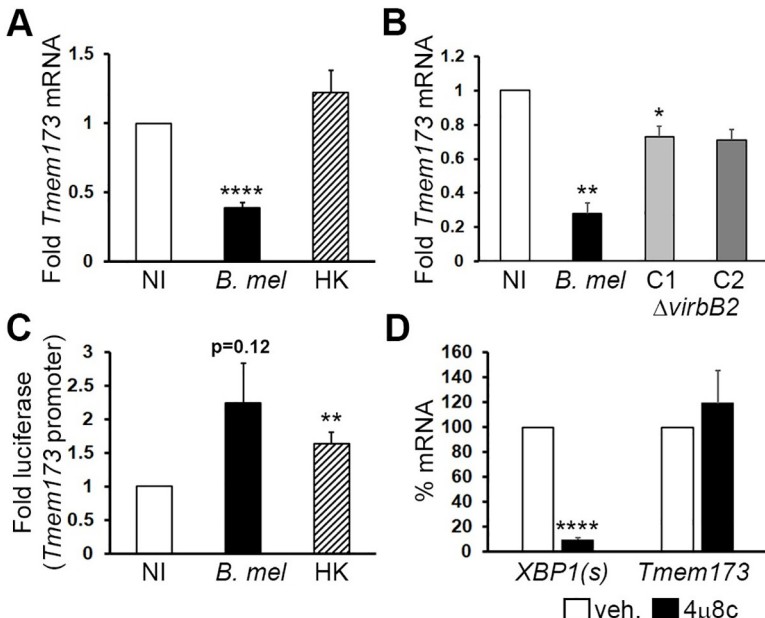

**Fig 3. *Brucella* down-regulation of STING requires live bacteria and Type IV secretion, and is RIDD-independent.** Macrophages were not infected (NI) or infected with 100 MOI *B. melitensis* 16M (*B. mel*), heat-killed *B. melitensis* (HK) in (A), or 2 clones of the Δ*virB2* mutant *B. melitensis* (C1 and C2) in (B) for 24 hours prior to harvesting for RNA processing. Relative *Tmem173* mRNA levels were determined by qPCR with normalization to 18S rRNA and uninfected controls (NI set = 1); p-values are vs. NI. Results are from 21 experiments for HK and 3 independent experiments for the VirB2 mutants. C) Macrophages were transfected with a murine *Tmem173* promoter luciferase reporter. Cells were then infected with 100 MOI live (*B. mel*) or heat-killed *B. melitensis* (HK). Lysates were analyzed by dual luciferase assay. Results are from 4 and 7 experiments respectively. P-values are vs. NI. D) Macrophages pre-treated with vehicle (veh.) or with the IRE1 endonuclease inhibitor 4μ8c one hour prior to infection with *B. melitensis* (N = 3 experiments). Levels of spliced XBP1 (XBP1(s)) or *Tmem173* mRNA were determined by qPCR. Vehicle treated mRNA expression was set = 100%.

Our group and others have previously shown that *Brucella* infection induces the Unfolded Protein Response (UPR) in macrophages [23,24]. An important effector of the UPR is the transmembrane protein Inositol-requiring enzyme 1 (IRE1), which functions as both a kinase and an endonuclease. Both *B. abortus* and *B. melitensis* infections activate the IRE1 pathway [23,25]. Following activation and oligomerization, the IRE1 endonuclease cleaves 26bp from the XBP1 transcription factor mRNA, thus removing a premature stop codon in the "spliced" product [26]. With prolonged endoplasmic reticulum (ER) stress, the IRE1 endonuclease changes activity to a process termed RIDD (Regulated IRE1 Dependent Decay), whereby it non-specifically degrades ER-proximal mRNAs in the secretory pathways, thus decreasing ER client load [27]. To determine if RIDD degrades *Tmem173* mRNA, macrophages were pre-treated with the IRE1 endonuclease inhibitor 4μ8c [28] before infection with *B. melitensis*. As a positive control for 4μ8c efficacy, we assessed inhibition of XBP1 splicing during *B. melitensis* infection via RT-qPCR (**Fig 3D**). *Tmem173* levels were unaffected by 4μ8c pre-treatment, indicating that STING mRNA down-regulation does not occur via IRE1-dependent endonuclease activity.

## *Brucella* infection upregulates miR-24, a STING-targeting microRNA

Another hypothesis for the reduction in STING mRNA is that its mRNA is a target of microRNA (miRNA). miRNA are endogenous, small non-coding RNAs 18–25 nucleotides in length that post-transcriptionally regulate gene expression via translational inhibition and mRNA

destruction [29]. To search for possible miRNAs that target STING, we used the online tool TargetScanMouse to identify possible candidates. A top hit was a conserved micro-RNA miR-24, which has been shown to post-transcriptionally regulate endogenous STING in *Rattus norvegicus* epithelium cells [30]. More recently, a study of liver ischemia reperfusion injury reported critical downregulation of STING via miR-24-3p [31]. MiR-24-2 (encoded by the *Mirn23a* locus) was also increased in our RNAseq data set from *Brucella*-infected macrophages [17]. To confirm the effect of infection on miR-24 levels, macrophages were infected for 24 hours with *B. melitensis* (**Fig 4A**). Infected macrophages significantly and reliably increased miR-24 levels compared to uninfected cells, although the degree of induction was variable (50% up to 8-fold). MiR-24 induction was also observed *in vivo*, 24h post-infection in mouse spleen. To confirm the biologic relevance of this miR-24 increase, we examined expression of another predicted mRNA target BCL2-like 11 (Bim), an apoptosis facilitator [32]. *Bcl2l11* mRNA levels were also significantly decreased in *B. melitensis* infected macrophages compared to uninfected cells (**Fig 4B**). *Tmem173* levels decreased over time as miR-24 increased (**Fig 4C**). Fold induction of miR-24 significantly correlated with the extent of *Tmem173* suppression (**Fig 4D**). Just as heat-killed *Brucella* failed to suppress STING, the killed *Brucella* did not induce miR-24 expression (**Fig 4E**). *Brucella* Δ*virB2* mutants also displayed a marked defect in miR-24 induction, consistent with the failure to fully suppress *Tmem173* expression (**Fig 4F**). These defects in miR-24 induction and *Tmem173* suppression were evident by 4 and 8h post infection, respectively (**Fig 4G**) and occurred in primary BMDM (**S1 Fig**). In the absence of VirB2, *Tmem173* expression decreased slightly at 4h, but then did not diminish further. One explanation for the lesser effects on host gene expression could be impaired survival of the

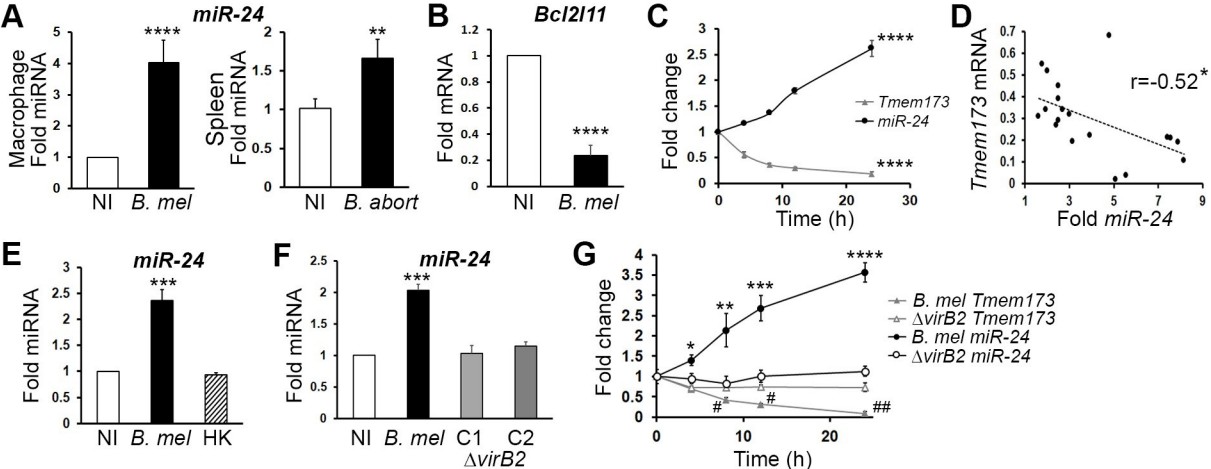

**Fig 4. *Brucella* induces a STING-targeting microRNA miR-24.** A) Left panel: Macrophages were not infected (NI) or infected with 100 MOI *Brucella melitensis* (*B. mel*) for 24 hours before harvesting for RNA. Right: Mice were infected with $10^6$ *B. abortus* 2308 *(B. abort)* for 24 hours prior to processing of spleen for microRNA. Micro RNA levels were determined by qPCR with normalization toRNU6 and uninfected controls (NI set = 1). *In vitro* results are from 17 experiments, with error bars denoting SEM. *In vivo*, results are representative of 2 independent experiments. N = 3 uninfected and 4 infected mice, with SD error bars. B) Macrophages were infected as in (A) and processed for mRNA. Expression was normalized to 18S rRNA. *Bcl2l11* (Bim) expression is from 8 experiments. C) Time course of quantitative expression of both miR24-3p and STING (*Tmem173*) mRNA. P<0.001 for changes over time (N = 6). D) Correlation is from 19 experiments performed and evaluated as in (A). $R^2 = 0.027$, p = 0.022 E) Macrophages were infected as in (A) and processed for miRNA. Comparison of live and heat killed *B. melitensis* (*B. mel* vs. HK) is from N = 9. P<0.005 for *B. mel* vs. NI and HK. F) Macrophages were infected with wild type *B. melitensis* or VirB2 deletion mutant clones C1 and C2 and analyzed as in (A). Results are from 3 experiments. P<0.005 for *B. mel* vs. NI and vs. ΔVirb2 clones. G) Time course comparing effects of wild type *B. melitensis* (filled symbols) and Δ *virB2* (clone 1, open symbols) on miR-24 (black circles) and *Tmem173* mRNA (gray triangles). Gene expression changes were normalized to time 0 for each *Brucella* genotype infection (see methods) and error bars represent standard deviations of triplicate determinations. P-values compare *Brucella* genotypes at each time point: #p<0.005, ##p<0.001 for *Tmem173* and *p<0.05, **p<0.01, ***p<0.005, ****p<0.001 for miR-24.

Δ*virB2* mutants. However, the defects in miR-24 induction and STING suppression were evident well before the Δ*virB2* mutant diverged from wild type in replication (**S1 Fig**). Host gene modulation defects at 8h were complemented with exogenous *virB2* (**S1 Fig**), confirming the gene specificity of the phenotype.

The requirement for live bacteria and the type IV secretion system to induce miR-24 and suppress STING expression suggested an active, bacterially driven process, rather than a passive host response to pathogen associated molecular patterns (PAMPs). Intriguingly, Ma et al. had reported that LPS suppressed STING expression via a MyD88-dependent pathway [33]. The mechanism was, and remains unknown. MyD88, a critical signaling intermediary downstream of multiple Toll-like receptors, is critical for control of *Brucella* infection *in vivo* [34,35]. To determine if MyD88 contributed to *Tmem173* downregulation in *Brucella*-infected macrophages, we compared *Tmem173* and miR-24 expression in *MyD88*⁻/⁻ and wild type macrophages (**Fig 5A and 5B**). MyD88 was not required for *Tmem173* mRNA suppression or for miR-24 induction, although both were less robust in *MyD88*⁻/⁻ cells. Further, miR-24 induction in STING⁻/⁻ macrophages is similar to wild type cells, indicating miR-24 induction does not require STING expression (**Fig 5C**). As another approach to examining whether *Brucella* PAMPS contribute to STING suppression, the regulation of *Tmem173* expression by purified Toll-like receptor (TLR) agonists was examined. *IL6* mRNA served as a control for stimulation. *Brucella* stimulates TLR2, TLR9 and TLR4, although *Brucella* LPS is 3–4 logs less endotoxic than *E. coli* LPS [35,36]. The ligands for TLR2 and TLR9, Pam3CSK4 and ODN 1585, respectively did not downregulate *Tmem173*, nor did purified *Brucella* LPS (**Fig 5D**).

To confirm that miR-24 is required for the down-regulation of STING and Bim, we utilized anti-miR-24 miRNA inhibitors (**S2 Fig**). The restoration of *Tmem173* and *Bcl2l11* expression with anti-miR-24 treatment (**Fig 6A**) was consistent with the idea that miR-24 contributes to the down-regulation of these mRNAs during *Brucella* infection. STING is required for optimal

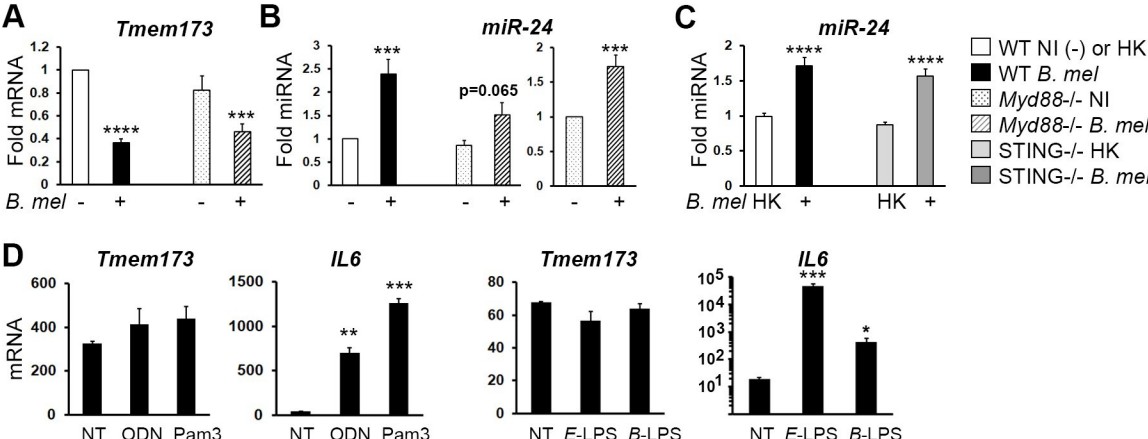

**Fig 5. MiR-24 induction and *Tmem173* suppression do not require STING or TLR agonist PAMPs.** A) Wild type (WT) or *Myd88*⁻/⁻ macrophages were infected with 100 MOI *B. melitensis* (*B. mel*) for 24h and then RNA levels assessed by qPCR as above. A) *Tmem173* expression is from N = 20, with normalization to WT uninfected controls within each experiment (WT NI = 1). B) MiR-24 expression is from N = 10, normalized as in (A) (left panel) and normalized to uninfected MyD88-/- (right). C) WT or STING (*Tmem173*)-/- macrophages were infected with heat killed (HK) or live *B. melitensis* for 24h and analyzed for miR-24 expression by qPCR, with normalization to uninfected controls (NI = 1), N = 8. White bars: uninfected or HK-infected wild type; black bars: infected wild type; dotted bars: uninfected *Myd88*⁻/⁻; striped bars: infected *Myd88*⁻/⁻; light gray: HK infected STING-/-; dark gray: live *Brucella* infected STING⁻/⁻. D) Macrophages were treated for 24h with media (not treated, NT), 1 μM ODN 1585 (ODN, TLR9 agonist), 10ng/mL Pam3CSK4 (Pam3, TLR2 agonist), 100ng/mL *E. coli* LPS (E-LPS, TLR4 agonist) or 10μg/mL *Brucella* LPS (B LPS), and analyzed for *Tmem173* and *IL6* expression. *E. coli* LPS is from 3 independent experiments, and ODN 1585, Pam3CSK4, and *Brucella* LPS data from 2 experiments. P-values are vs. NT.

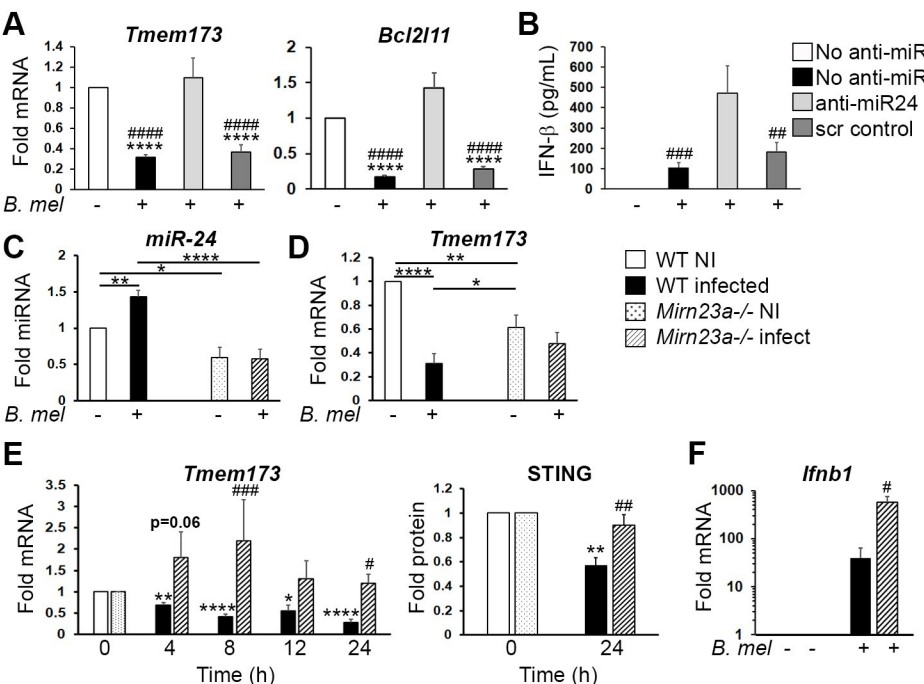

**Fig 6. STING suppression requires miR-24 induction.** A) Macrophage cells were transfected with an anti-miR24 inhibitor or control anti-miR, then infected with 100 MOI *B. melitensis* (*B. mel*) for 24h. Relative gene expression of *Tmem173* (left) and *Bcl2l11* (right) were determined via qPCR with normalization to 18S rRNA and non-infected control (NI set = 1). N = 5 (*Tmem173*) and 3 experiments (*Bcl2l11*). P-values are vs. NI (*) or vs. anti-miR24 (#). (A and B) White bars are NI; black *B. mel*; light gray anti-miR-24 + *B. mel*; and dark gray scrambled anti-miR control + *B. mel*. B) IFN-β production in culture supernatant after 24h of infection was determined by ELISA. Data are from 4 experiments. C) And (D) Wild type (WT) or *Mirn23a*$^{-/-}$ macrophage cells were infected with 100 MOI *B. melitensis* for 24h and expression of miR-24 (C) *Tmem173* mRNA (D) determined as above. Results were normalized to uninfected wild type (NI = 1) within each experiment. Data are from 5 and 8 experiments respectively. E) WT or *Mirn23a*$^{-/-}$ macrophages were infected with *B. melitensis* for the times indicated prior to lysis for RNA extraction. *Tmem173* levels were determined using qPCR with each genotype normalized to its own NI values (set = 1). 24h data is from 9 experiments with the other time points assessed in 5 experiments. STING protein 24h following infection was detected using western blot with normalization to β-actin and genotype-respective uninfected controls. N = 3. In E, p-values are for *B. mel* infected vs NI WT cells (*) and for WT vs *Mirn23a*$^{-/-}$ infected cells (#). F) *Ifnb1* mRNA expression at 24h, normalized to uninfected control cells for each genotype. N = 3, #p-value is for WT vs *Mirn23a*-/-. For C-F, dotted bars are uninfected *Mirn23a*$^{-/-}$ and striped bars are infected *Mirn23a*$^{-/-}$ cells.

*Brucella*-dependent IFN-β production in macrophages [9,18]. To determine if failure to suppress STING correlated with increased STING activity, we assessed the impact of the anti-miR-24 on IFN-β production. As shown in **Fig 6B**, IFN-β was significantly up-regulated in macrophages transfected with the miR-24 inhibitor compared to mock transfected control cells, consistent with increased STING activity. Together, these data support the idea that *Brucella* infection induces miR-24 to down-regulate STING.

To further evaluate the requirement for miR-24, we utilized a genetic model of miR-24 deficiency. MiR-24-3p is 100% homologous between mouse, rat and human and is expressed from two genetic loci: *Mirn23a* encodes miR-23a, miR-24-2 and miR-27a and *Mirn23b* encodes miR-23b, miR-24-1 and miR-27b. Our previous RNAseq data suggested bone marrow macrophages induced miR-24-2 but not miR-24-1 [17]. *Mirn23a* is the predominant source of miR-24 in blood [37]. *Mirn23a*$^{-/-}$ macrophages showed decreased levels of miR-24 compared to wild type prior to infection and were deficient at miR-24 upregulation in response to *Brucella* infection (**Fig 6C**). As noted above, heat-killed *Brucella* did not induce miR-24 in either genotype. *Mirn23a*$^{-/-}$ macrophages were unable to suppress *Tmem173* expression at 24h in relation

to their uninfected state, although overall levels of *Tmem173 mRNA* were decreased compared to uninfected wild type macrophages, suggesting a balance between static miR-24 and *Tmem173* levels (**Fig 6D and 6E**). STING protein suppression was also impaired. The defect in *Tmem173* suppression in the *Mirn23a*[-/-] macrophages, more evident over time (**Fig 6E**), correlated with greatly increased *Ifnb1* induction by 24h post-infection, consistent with increased STING activity (**Fig 6F**).

## Decreased *Brucella* replication in *miR23* locus-/- macrophages

Although the data in **Fig 1** suggested that STING regulates *Brucella* infection, the biologic consequences of miR-24 induction and STING suppression were not clear. To determine the role of the *Mirn23a* locus in infection, we compared replication (CFU) in wild type vs. *Mirn23a*[-/-] macrophages (**Fig 7A**). Initial uptake of *Brucella* was similar between genotypes, but diverged by 8h, with lower *Brucella* CFU recovered in the *Mirn23a*[-/-] macrophages. This divergence maintained or increased over the course of infection through 48–72 hours. These results were consistent with a role for the miRs encoded by this locus in supporting intracellular infection. We further confirmed that the observations *in vitro* were relevant *in vivo* by infecting *Mirn23a*-/- mice with *Brucella*. As predicted by the *in vitro* data, *Mirn23a*-/- mice were more resistant to *B. neotomae*, with greater than 1 log less splenic CFU one week post-infection. For gene expression in these splenocytes, see **S3 Fig**. To confirm the specificity of the *Mirn23a*-/- phenotype for miR-24, anti-miR24 and miR-24 mimics (**Fig 7B**) were introduced. Anti-miR24 greatly decreased the capacity of wild type but not *Mirn23a*[-/-] macrophages to control

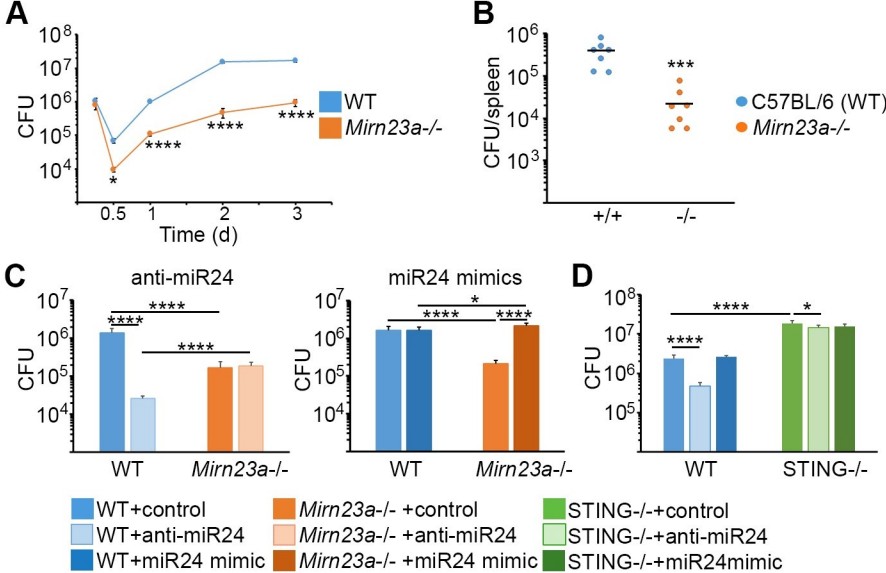

**Fig 7. Failure to induce miR-24 inhibits *Brucella* replication.** A) Wild type (WT, blue symbols) or *Mirn23a*-/- macrophages (orange symbols) were infected with 100 MOI *B. melitensis* for the times indicated, lysed, and then CFU were enumerated. Error bars are standard deviations of 8 replicates and results are representative of 3 independent experiments. B) C57BL/6 or *Mirn23a*-/- mice were infected with $10^6$ *B. neotomae* (N = 7 mice per group). After 7 days, spleens were harvested for analysis of CFU. Bars are mean values. C) WT or *Mirn23a*[-/-] macrophages were transfected with anti-miR24 (left panel) or miR-24-3p mimic (right) or miR control then infected with *B. melitensis* for 24h. Cells were then lysed and CFU enumerated. Error bars are standard deviations of 8 replicates and representative of 4–5 experiments for the anti-miRs and miR-24-3p mimics respectively. Paler bars represent transfection of anti-miR24, whereas darker bars represent addition of the mimic. D) WT or STING[-/-] macrophages were transfected with anti-miR24 or miR-24-3p mimics and then infected with 100 MOI *B. melitensis* for 24h prior to enumeration of CFU. Results are representative of 4 experiments.

intracellular *B. melitensis* replication. In the converse experiment, addition of miR-24 mimics significantly enhanced *B. melitensis* replication in the *Mirn23a*$^{-/-}$ but not always in wild type macrophages. These data were consistent with the hypothesis that miR-24 is responsible for the decreased replication in *Mirn23a*$^{-/-}$ cells. Finally, to determine what proportion of the miR-24 effect was due to STING (vs. other miR-24 targets), STING$^{-/-}$ macrophages were transfected with anti-miR24 or mimics prior to infection. Whereas anti-miR24 suppressed *Brucella* replication in wild type macrophages, neither mimics nor anti-miRs exerted a significant magnitude of effect on replication in STING$^{-/-}$ cells (**Fig 7C**, anti-miRs: 5-40-fold in wild type vs <1.5-fold difference in STING$^{-/-}$). These epistasis results suggested STING accounts for the majority of the miR-24 effect on replication during infection. Together, these data are consistent with the hypothesis that *Brucella* induction of miR-24 suppresses STING expression to increase infectious success.

## Discussion

The cytosolic DNA sensor STING plays a key role in innate immune defense via transcription of host defense genes including Type I interferons, induction of NF-κB-dependent responses and autophagy [15]. *B. abortus* DNA and cyclic-di-GMP activate STING, triggering IFN-β production [9,18]. In this report, at one, 3 and 6 weeks post infection, STING$^{-/-}$ mice had a 1-2-log higher burden of *Brucella* compared to age-matched wild-type counterparts, confirming that STING is ultimately required for control of *Brucella* infection. Although STING protects against *Brucella*, the striking suppression of STING mRNA expression early following infection suggests *Brucella* actively sabotages this innate immune sensor to gain a foothold inside macrophages.

STING suppression occurred post-transcriptionally, independently of UPR-mediated RNA decay, via upregulation of the microRNA miR-24. MiR-24 induction and STING suppression required live bacteria, and full suppression required the VirB-encoded Type IV secretion system, suggesting an active bacterial driven process rather than a simple host response to *Brucella* PAMPs. The independence of miR-24 upregulation from STING signaling and lack of STING suppression by purified TLR agonists supports this model. MiR-24 upregulation and STING downregulation were slightly less robust in the MyD88-/- macrophages, consistent with a minor role for MyD88 signaling. Divergence of the Δ*virB2* mutant and wild type *Brucella* <8h post-infection suggests the requirement for the type IV secretion system does not simply reflect its effect on replication. Rather, the type IV secretion system may contribute to miR-24 upregulation through secretion of a specific *Brucella* substrate or by enabling appropriate intracellular trafficking. The early divergence (4h) of STING expression between *Mirn23a*$^{-/-}$ and wild type macrophages supports the idea that the effect of miR-24 precedes intracellular *Brucella* replication.

Induction of the *Mirn23a* gene locus during infection comes at potential cost for *Brucella* infection. In NK cells, deletion of this locus (also known as *Mirc11*) resulted in decreased ability to contain *Listeria* infection, related to diminished IFN-γ and pro-inflammatory cytokine production [38]. Both IFN-γ and TNF-α have long been known to be critical for control of *Brucella* infection. However, effects of miR-24 on cytokine production may be cell-type specific. In CD4+ T cells, miR-24 was reported to target IFN-γ mRNA [39]. Over-expression of miR-24 in a *Staphylococcus aureus* infection model decreased "M1" inflammatory mediator production in macrophages and enhanced "M2" marker expression, which would benefit *Brucella* [40,41]. An earlier study had also suggested miR-24 modulates macrophage polarization towards an "alternative" M2 phenotype [42]. Manipulation of miR-24 levels had minimal effects on *Brucella* replication in STING$^{-/-}$ cells, suggesting that STING is the dominant or

primary target of miR-24 induction during *Brucella* infection of macrophages that impacts intracellular replication.

Recently, the ability of another chronic intracellular pathogen, *Mycobacterium tuberculosis*, to manipulate host innate responses, autophagy, and apoptosis via host miRNA has garnered much interest [43–46]. In contrast, there is much less information regarding miRNA in the context of *Brucella* infection [47–49]. Budak et al. investigated the miRNA expression patterns in CD4+ and CD8+ T cells from patients, and reported discrete changes with acute vs. chronic brucellosis [50,51]. Another study reported the up-regulation of miR-1981 in RAW264.7 infected macrophages and showed the interaction of that microRNA with the 3'-UTR of Bcl-2, an apoptosis regulator [52]. Recently, Corsetti et al revealed several miRNA-dependent mechanisms of immune manipulation during *B. abortus* infection: upregulation of mmu-miR-181a-5p suppressed TNF-α and miR-21a-5p downregulation decreased IL10 and elevated GBP5 [49]. Here, we show that *Brucella* significantly induce miR-24. Additionally, another predicted target of miR-24, Bim, a key apoptosis-regulator induced by PERK signaling and C/EBP homologous protein (CHOP) transcriptional activity, was significantly down regulated during infection with *B. melitensis*. MiR-24 inhibition resulted in a significant recovery in both Bim and STING, indicating that miR-24 is targeting both these mRNAs during *B. melitensis* infection.

Our previous RNAseq data set [17] revealed upregulation of miR-24-2, encoded at the *Mirn23a* locus, but not miR-24-1 from the *Mirn23b* locus. Furthermore, although *Mirn23a*[-/-] cells expressed some miR-24, they were unable to upregulate its expression in response to infection. Throughout these experiments, fold-induction of miR-24 correlated with STING mRNA suppression. These results suggest that upsetting the balance between miR-24 and *Tmem173* levels is the critical component. The strong effects of miR-24 manipulation through mimics and anti-miRs, as well as the defects in replication in the *Mirn23a*[-/-] macrophages together support the idea that upregulation of miR-24 is important for replication early during infection. *In vivo*, the inability to upregulate miR-24 correlated with decreased splenic CFU in the *Mirn23a-/-* mice one week post-infection, also suggesting that miR-24 supports acute infection. The greater replication in the *Mirn23a*[-/-] cells vs the anti-miR24 treated wild type macrophages (**Fig 7**) may reflect contributions from miR-23a and miR-27a encoded by that locus.

In addition to the increased STING mRNA, miR-24 inhibition or genetic deficiency resulted in a significantly increased IFN-β response compared to uninhibited macrophages. Although initially identified in its role in viral protection, Type I interferons have recently become a topic of interest in response to many bacterial pathogens [53]. During infections, the effect of Type I interferons can be protective or detrimental depending on the bacterial species. For example, Type I interferon protects mice against *Salmonella typhimurium* infection whereas Interferon-alpha/beta receptor (IFNAR)-mediated Type I interferon responses to *Francisella tularensis* and *Listeria monocytogenes* are harmful to the host [54–56]. The role of Type I interferon in response to *Brucella* is currently unclear; a study in 2007 showed no difference in splenic and liver CFUs in wild type versus *IFNAR*[-/-] mice [57]. However, a more recent study has shown a higher burden of *Brucella* in wild type mice compared to *IFNAR*[-/-] mice, indicating that Type I interferon response is detrimental to the host [9]. Resistance to *B. abortus* in the *IFNAR*[-/-] mice was accompanied by elevated production of IFN-γ and NO, and decreased apoptosis compared to wild-type mice. Although type I IFN served as a useful indicator for STING activity in our study, ultimately, the experience with the *IFNAR*[-/-] mice suggest STING is controlling infection through Type I IFN-independent mechanisms.

*Brucella* potently inhibits apoptosis, contributing to chronic infection; however, the mechanisms behind this process are unknown [34,58]. By down-regulating STING and subsequent

IFN-β production, *Brucella* could be actively inhibiting apoptosis that is dependent upon Type I IFN signaling. Further, by up-regulating miR-24, which in turn down-regulates Bim, *Brucella* could be avoiding UPR-mediated apoptosis, which is partially dependent upon Bim in other experimental systems [59]. *B. melitensis* infection robustly induces CHOP, which is an upstream activator of Bim [23,60]. We did not detect a reliable effect of the anti-miR-24 on host cell apoptosis or cell death. One likely explanation is that there are unidentified miR-24-independent mechanisms that inhibit apoptosis independently of STING and Bim down-regulation. Indeed, our previous RNAseq data [17] suggested that *Brucella* suppresses the expression of multiple pro-apoptotic molecules.

In summary, our findings document the evasion of full STING activation during infection by an intracellular bacteria pathogen via miR-24-mediated suppression of STING expression. It is noteworthy that a single miRNA species should have such a profound impact on a major cytosolic innate immune sensor and consequent *Brucella* replication. Our data may have implications for other important pathogens. For instance, miR-24 was up-regulated and cited as one of 7 significantly altered microRNAs controlling the transcriptional response to *M. tuberculosis* in macrophages [61]. In a separate report, in transcriptomic data, *Tmem173* was suppressed by more than 50% at 4 hours and >75% decreased 12 hours following *M. tuberculosis* infection [62]. Widely considered a "stealth" pathogen, *Brucella* can evade immune surveillance and persist chronically in macrophages [63]. In contrast to this idea of *Brucella* as "flying under the radar", previous reports have described *Brucella* subversion of toll-like receptor signaling via Btp1/TcpB [64–66]. The data presented here elucidate a critical mechanism by which *Brucella* actively sabotages cytosolic surveillance by the innate immune sensor STING to establish its intracellular niche.

## Methods

### Ethics statement

Mouse care, handling and experimental procedures were approved by the Institutional Animal Care and Use Committees (IACUC) of the institutions involved in this project and performed with strict adherence.

Reagents, resources and associated sources and identifiers are listed below in Table 1 and primers in Table 2.

### Experimental model and method details

**Mice.** *In vivo* infections in mice were performed via intraperitoneal injection of $10^6$ CFU/0.2ml of PBS diluent of *Brucella abortus* 2308 or *Brucella neotomae* Stoenner and Lackman 1957 or 0.2ml PBS (uninfected vehicle control). At indicated time points post infection, mice were euthanized following IACUC approved procedures. In Brazil, wild type C57BL/6 mice were purchased from the Federal University of Minas Gerais (UFMG). STING (*Tmem173*)$^{-/-}$, mice were described previously [13]. Mice were maintained at UFMG and used at six weeks of age. To count *Brucella* colony forming units (CFU), individual spleens were macerated in 10 ml saline, serially diluted, and plated in duplicate on *Brucella* Broth agar. After 3 days of incubation at 37˚C, the number of CFU was determined as described previously [34]. At the University of Notre Dame, seven wild-type C57/BL6 and *Mirn23a-/-* mice (mixed gender; 6 wks) were infected with *B. neotomae*, and one per genotype injected with PBS control, and euthanized at 7 days post-infection. Spleens were processed and assayed for CFU and RNA as follows: Single cell suspensions of mouse spleens were prepared using a gentleMACS dissociator (Miltenyi Biotec) following the manufacturer's protocol. Splenocytes from each mouse were processed for either CFU determination (as described below) or RNA preparation.

**Table 1. Reagents and resources used in this study and their associated sources and identifiers.**

| REAGENT or RESOURCE | SOURCE | IDENTIFIER |
|---|---|---|
| **Antibodies** | | |
| Rabbit monoclonal anti-STING | Cell Signaling Technology | Cat# 13647 |
| Mouse monoclonal anti-β-actin | Santa Cruz | sc-47778 |
| **Bacterial Strains** | | |
| *Brucella melitensis 16M* | UW-Madison archive | N/A |
| *Brucella abortus 2308* | UW-Madison archive | N/A |
| *Brucella neotomae* Stoenner and Lackman 1957 | ATC 23459 | N/A |
| *Brucella suis 1330* | UW-Madison archive | N/A |
| *Escherichia coli DH5a* | UW-Madison archive | N/A |
| **Chemicals** | | |
| 4μ8c | EMD Millipore | Cat#412512-25MG |
| ODN 1585 | Invivogen | Tlrl-1585 |
| Pam3CSK4 | Invivogen | Tlrl-pms |
| **Critical Commercial Assays** | | |
| RNAzol RT | Molecular Research Center, Inc. | RN 190 |
| PowerUp SYBR Green Master Mix | AppliedBiosystems | Cat# A25742 |
| B-PER Bacterial Protein Extraction Reagent | ThermoFisher Scientific | Cat# 78248 |
| M-PER Mammalian Protein Extraction Reagent | ThermoFisher Scientific | Cat# 78501 |
| Halt Protease and Phosphatase Inhibitor | ThermoFisher Scientific | Cat# 78440 |
| Dual-Glo Luciferase assay system | Promega | Cat# E2490 |
| Cell Line Nucleofector kit V | Lonza | VVCA-1003 |
| Lipofectamine RNAiMAX Transfection Reagent | ThermoFisher Scientific | Cat# 13778030 |
| Opti-MEM | ThermoFisher Scientific | Cat# 31985070 |
| LEGEND MAX Mouse IFN-β ELISA Kit with Pre-coated Plates | BioLegend | Cat# 439407 |
| qScript microRNA cDNA Synthesis Kit | Quanta Biosciences | Cat# 95107–025 |
| SuperScript IV VILO Master Mix | ThermoFisher Scientific | Cat# 11756050 |
| LPS Extraction Kit | iNtRON Biotechnology | Cat# 17141 |
| **Experimental Models: Cell Lines** | | |
| iMACs (immortalized macrophages) | John-Demian Sauer | N/A |
| LADMAC (for CSF-1) | ATCC | CRL-2420 |
| **Experimental Models: Organisms/Strains** | | |
| *Mus musculus* C57BL/6 | Federal University of Minas Gerais; IUSM South Bend | N/A |
| *Mus musculus* C57BL/6 STING-/- | Ishikawa and Barber, 2008 | N/A |
| *Mus musculus mirn23a-/-(on C57BL/6 background)* | Richard Dahl; IUSM South Bend | N/A |
| *Mus musculus MyD88-/-(on C57BL/6 background)* | Bruce Klein | N/A |
| **Oligonucleotides and plasmids** | | |
| mirVana miRNA inhibitor hsa-miR-24-3p | ThermoFisher Scientific | Cat# 4464084 |
| Anti-miR miRNA Inhibitor Negative Control #1 | ThermoFisher Scientific | Cat# AM17010 |
| miRNA Neg Control | SIGMA | MISSION miRNA Control |
| miRNA hsa-mir-24 | SIGMA | MISSION miRNA Mimic |
| miRNA inhibitor mmu-mir-24-3p | SIGMA | MISSION miRNA Inhibitor |
| VirB2 knock-out vector, pAV2.2 | Renee M. Tsolis | Den Hartigh et al., 2004 [22] |
| pBBR1-MCS4 | Lab Archive | N/A |
| pBBR1-VirB2 | This work | N/A |
| pSTING-254 murine STING promoter luciferase plasmid | Hua-Guo Xu | Xu et al., 2017 [67] |
| **Software and Algorithms** | | |
| GraphPad Prism 7 | Graphpad Software | N/A |

**Table 2. Primers used in this study.**

| Primers used in this work | | |
|---|---|---|
| Primer | Sequence | Source |
| 18S F | AGGGGAGAGCGGGTAAGAGA | IDT |
| 18S R | GGACAGGACTAGGCGGAACA | IDT |
| Bim F | TGTCTGACTCTGATTCTCGGA | IDT |
| Bim R | TGCAATTGTCCACCTTCTCTG | IDT |
| hsa-miR-24-3p | NA | SIGMA; MIRAP00056 |
| IFN-β F | GGCATCAACTGACAGGTCTT | IDT |
| IFN-β R | ACTCATGAAGTACAACAGCTACG | IDT |
| RNU6 | GCAAATTCGTGAAGCGTTCC | IDT |
| STING F | AAGTCTCTGCAGTCTGTGAAG | IDT |
| STING R | TGTAGCTGATTGAACATTCGGA | IDT |
| Xbp-1(t) F | TCCGCAGCACTCAGACTATGT | IDT |
| Xbp-1(t) R | ATGCCCAAAAGGATATCAGACTC | IDT |
| Xbp-1(s) F | GAGTCCGCAGCAGGTG | IDT |
| Xbp-1(s) R | GTGTCAGAGTCCATGGGA | IDT |
| virB2 F | CCAGACCGATAAGAGAACGATG | IDT |
| virB2 R | CCGATCAGGCACGCATATAA | IDT |
| virB2-988 F | CTCGAGGCTGCCCCAGTAAAAAAAACGAC | IDT |
| VIRB2-1562 R | ATCGATTCGGTCTGCTTGCTCAATGTCTAT | IDT |
| PerfeCTa Universal PCR Primer | NA | Quanta Biosciences; 95109–500 |

Primers used for qPCR determination of gene expression and their sources are listed.

**Mammalian cell lines.** *V-raf/v-myc* immortalized murine bone marrow derived macrophages (BMDM) were a generous gift from Dr. John-Demian Sauer at the University of Wisconsin-Madison. These macrophages (iMacs) were from C57BL/6 mice. *V-raf/V-myc* immortalized BMDM were generated in our lab from leg bones of *MyD88*$^{-/-}$, STING$^{-/-}$, and *Mirn23a*$^{-/-}$ mice obtained from researchers listed in Table 1 above. These mice were all on a C57BL/6 background. For the *Mirn23a-/-* experiments, wild type C57BL/6 bones were obtained from Dr. Dahl's colony. All immortalized macrophage cell lines were cultured at 37˚C with 5% CO$_2$ in RPMI supplemented with 1mM Na pyruvate, 0.05mM 2-mercaptoethanol, and 10% FBS. Apart from Figs 1 and S1, immortalized macrophages were used for *in vitro* experiments.

**Bone marrow derived macrophages (BMDM).** To generate CSF-1 containing media, LADMAC cells were grown to confluency and then pelleted. The supernatant was filtered (0.45 μm) prior to use. Mouse femur and tibia bones were cleaned of tissue. The ends of the bones were snipped aseptically using surgical scissors, and bone marrow was flushed using a PBS-filled syringe with a 27G needle. The flushed cells were then pulled and ejected through a syringe with an 18G needle to generate a single cell suspension. The suspension was filtered through a 70 μm cell strainer to remove solid fragments, pelleted (300 x g; 5 min), and resuspended in 35% LADMAC conditioned media. Bone marrow cells were differentiated into macrophages over 7 days.

**Brucella strains.** *Brucella* strains *B. melitensis*, *B. abortus*, *B. suis*, and *B. neotomae* were from archived stock of the University of Wisconsin-Madison and the University of Minas Gerais (*B. abortus* used in Figs 1 and 4). *Brucella* were cultured using Brain Heart Infusion broth or agar (Difco) at 37˚C. All experiments with select agent *Brucella* strains were performed in a Biosafety Level 3 facility in compliance with the CDC Division of Select Agents and Toxins regulations according to standard operating procedures approved by the University of

Wisconsin-Madison and UFMG Institutional Biosafety Committees. *B. neotomae in vivo* experiments were approved by the University of Notre Dame Institutional Biosafety Committee.

VirB2 deletion mutants of *Brucella* were derived through homologous recombination following the methods described by den Hartigh et. al., 2004 using the plasmid pAV2.2 (generous gift from R.M. Tsolis). Briefly, exponentially growing *Brucella* were made electrocompetent following standard microbiological methods. Electrocompetent *Brucella* were then electroporated with pAV2.2 and VirB2 deletion mutants were selected for kanamycin resistance and carbenicillin sensitivity. VirB2 deletion was confirmed in these clones using PCR. Where not otherwise specified, "clone 1" was used in experiments. The complementation plasmid for VirB2 was engineered through PCR amplification of the VirB2 ORF plus the ribosome binding site but lacking the promoter sequences. Primers virb2-988 F and virb2-1562 R also encoded appropriate restriction sites to aid cloning (*Xho* I/*Cla* I). VirB2 was then cloned into pBBR1-MCS4 containing a constitutive *lac* promoter shown in previous studies to be effective for complementation analysis in *B. melitensis* [68]. The PCR product was digested and ligated with similarly digested pBBR1-MCS4 to generate the complementation plasmid. The VirB2 amplified product was directionally cloned into pBBR1-MCS4 to ensure that these genes were transcribed from the *lac* promoter present in the plasmid. The resulting complementation plasmid, pBBR1-VirB2, was transformed into the VirB2 deletion mutant and selected for ampicillin resistance. In experiments using heat-killed (HK) *Brucella* as a control, inactivation of *Brucella* was as follows. After growing *Brucella* in culture, the sample was quantitated using spectrophotometry. *Brucella* was then aliquoted in microcentrifuge tubes and placed in a 56˚C waterbath for 1 h. *Brucella* lipopolysaccharide (LPS) along with *E. coli* LPS was used as an agonist for TLR studies and was extracted as follows: *Brucella melitensis* was cultured in BHI broth for 3 days in a 37˚C shaking incubator to an $OD_{600}$ of approximately 1.2. Bacteria was pelleted and LPS extracted using an LPS extraction kit (iNtRon Biotechnology) following the manufacturer's protocol. *E. coli* DH5a LPS was extracted using the same method after 1 day of culture.

**Infections, treatments, transfections and CFU assays.** Immortalized macrophage cell lines were plated on 6-well tissue culture plates at $0.4 \times 10^6$ cells per well in 2 ml culture media. Primary BMDM were plated on 6-well tissue culture plates at $1.2 \times 10^6$ cells per well in 2 ml culture media. The next day, the media was replaced, and cells were infected with 100 MOI *Brucella* determined by spectrophotometry (OD at 600 nm) through a formula established by a *Brucella* growth curve. Cells were then centrifuged (300 x g; 5 min) to synchronize infection and were incubated at 37˚C with 5% $CO_2$. One hour later, cells were washed 3x with 2 ml/well warm PBS and fresh media containing 10 μg/ml gentamycin was added. This one-hour point was considered "Time 0" post infection for the time courses. Cell treatments: 4μ8c (IRE1 endonuclease inhibitor) was dissolved at 10mM in DMSO and then diluted to 1mM (100x) in media. 4μ8c was added to the cultures 1 hour prior to infection. For TLR agonist treatment, cells were plated in 12 well plates at $0.5 \times 10^6$/well in 1 mL growth media. ODN 1585 and Pam3CSK4 were dissolved in media and used at concentrations of 1 μM and 10ng/mL respectively. Extracted *E. coli* LPS was used at 100ng/mL and *Brucella* LPS at 10μg/mL. Luciferase assay: 2 μg STING promoter driven luciferase reporter plasmid and 0.5μg Renilla TK were transfected into $1 \times 10^6$ cells suspended in solution V by AMAXA Nucleofection (Lonza), then cells were plated in 12 well plates. The following day, the cells were given fresh media and infected with *Brucella melitensis* (100 MOI) or stimulated with heat-killed *Brucella melitensis* (100 MOI). Twenty-four hours later, the cells were washed with 1ml PBS 3x, resuspended in 150 μL kit lysis buffer and assayed according to the manufacturer's instructions. Assays were performed in triplicate using a Veritas microplate luminometer (Turner Biosystems). For microRNA transfections, immortalized macrophages were seeded on 6-well tissue culture plates at $0.4 \times 10^6$ cells/well. MiRNA mimics (miR-24-3p), miRNA control, and miRNA

inhibitors for miR-24-3p were diluted to 0.28 μM using Opti-MEM and the cells were transfected using using RNAiMAX reagent following the manufacturer's protocol. One day after transfection, cells were infected as described above, then processed for RNA or CFU. For CFU assays, cells were washed 3x in PBS to remove extracellular bacteria, then 1 ml of cell lysis buffer (dH$_2$O + 0.1% Triton X-100) was added per well. CFU were determined by serial dilution plating with 8 replicates on BHI agar after 3–4 days.

**Quantitative polymerase chain reaction (qPCR).** Total cellular RNA was processed using RNAzol RT reagent (Molecular Research Center, Inc.) following the manufacturer's protocol. Then cDNA was prepared from either mRNA, using the Superscript IV VILO system (Invitrogen) or microRNA using the qScript system (Quanta biosciences). Samples for Quantitative PCRs were analyzed via SYBR Green and the delta-delta Ct method to calculate relative fold gene expression using a StepOnePlus thermocycler (ABI). Endogenous housekeeping genes used for comparative expression were either 18S (rRNA) or RNU6 (microRNA). The primers used in this study were designed using IDT's online primer design tool or purchased.

**Quantification of IFN-β by ELISA.** Culture supernatants from cells were collected and frozen at -80˚C until assayed. A mouse IFN-β ELISA was performed following the kit manufacturer's protocol. Absorbance at 450 nm and 570 nm were determined utilizing a BioTek microplate reader. Mouse IFN-β was quantified by standard curve.

**Western blot assays.** Cell lines (infected or control) were washed with PBS, scraped off the well, then transferred to a microcentrofuge tube and pelleted (4k RPM, 5 min). The supernatant was removed and cells lysed with M-PER reagent (ThermoFisher Scientific) according to the manufacturer's protocol. Whole-cell lysates were resolved by 12% SDS-PAGE. Samples were then transferred to polyvinyldene difluoride (PVD) membrane and immunoblotted with anti-STING primary antibody (ThermoFisher Scientific) and anti-β-actin primary antibody (Santa Cruz), followed by a fluorescence-conjugated secondary antibody (LI-COR). Proteins were visualized and quantitated with the Odyssey system (LI-COR).

**Quantification and statistical analysis.** CFU values and standardized mRNA expression levels were summarized in terms of means ± standard deviations and displayed in graphical format using bar charts, stratified by experimental conditions. Comparisons between experimental groups were conducted using two-sample t-test or analysis of variance (ANOVA). Pairwise comparisons between multiple groups were conducted using Tukey's Honestly Significance Difference (HSD) method. Residual and normal probability plots were examined to verify the model assumptions. Linear regression and Pearson's correlation analyses were conducted to evaluate bivariate associations. Statistical significance is indicated in the figures (* $p<0.05$, ** $p<0.01$, *** $p<0.005$, **** $p<0.001$, ns not significant). Statistical analyses were conducted using SAS (SAS Institute Inc., Cary NC), version 9.4.

## Supporting information

**S1 Fig. The type IV secretion system is required for miR-24 upregulation and complete *Tmem173* downregulation.** A) Bone marrow derived macrophages were infected with 100 MOI *B. melitensis* or Δ*virB2* mutant and harvested for RNA at 12h and 24h post infection. *Tmem173* and *miR-24* expression was normalized to Time 0 for each genotype (set = 1). B) Immortalized macrophages were infected as above and CFU (8 replicates) determined over time. By 24h, the Δ*virB2* mutant was at a significant disadvantage for replication ($p<0.001$) vs. wild type *B. mel* and the complemented mutant. C) Macrophages were infected with 100 MOI of *B. melitensis*, Δ*virB2* mutant, or the mutant transfected with the *virB2* gene for 8h prior to harvest for RNA. Error bars are triplicate standard deviations.
(TIF)

**S2 Fig. miR-24 and STING expression in anti-miR-24 and miR-24 mimic transfected cells.** A) Macrophages were not transfected (white and black bars) or transfected with anti-miR-24 inhibitor (pale gray), scrambled control nucleotide (dark gray), or miR-24 mimic (striped bars). Cells were subsequently uninfected (-) or infected with 100 MOI *B. melitensis* (+). After 24h, cells were harvested for RNA and miR-24 levels determined by qPCR with normalization to RNU6. N = 3 experiments. Error bars are SEM. *p<0.05, ****p<0.001. B) Macrophages were transfected as in (A). 24h following infection with *Brucella*, cells were lysed and whole cell lysates resolved by SDS-PAGE. STING and β-actin were detected using western blot and immunofluorescence. Blot is representative of 2 experiments. Graph bars are actin-normalized optical densities of the STING bands.
(TIF)

**S3 Fig. *Mirn23a-/-* mice fail to upregulate miR-24 and exhibit less *Tmem173* suppression following *Brucella* infection.** C57BL/6 (+/+) and *Mirn23a-/-* (-/-) mice were infected intra-peritoneally with $10^6$ *B. neotomae* (7 mice per group). After 7 days, mice were sacrificed and spleens harvested for CFU (Fig 7) and RNA. Gene expression was determined by qPCR. Results from individual infected mice were normalized to an uninfected control (set = 1) for each genotype. ****p<0.001 and **p<0.01 vs. infected C57BL/6.
(TIF)

## Acknowledgments

We would like to thank Dr. John-Demian Sauer for providing us with the *v-raf/v-myc* immortalized murine bone marrow-derived macrophages and Glen Barber for providing the STING$^{-/-}$ mice. Bruce Klein supplied MyD88-/- femurs and Richard Dahl the *Mirn23a -/-* mouse bones.

## Author Contributions

**Conceptualization:** Mike Khan, Jerome S. Harms, Sergio Costa Oliveira, Richard Dahl, Glen N. Barber, Gary A. Splitter, Judith A. Smith.

**Formal analysis:** Jens Eickhoff.

**Funding acquisition:** Mike Khan, Sergio Costa Oliveira, Richard Dahl, Glen N. Barber, Gary A. Splitter, Judith A. Smith.

**Investigation:** Mike Khan, Jerome S. Harms, Yiping Liu, Jens Eickhoff, Jin Wen Tan, Tony Hu, Fengwei Cai, Erika Guimaraes, Richard Dahl, Yong Cheng, Delia Gutman.

**Methodology:** Mike Khan, Jerome S. Harms, Jens Eickhoff, Sergio Costa Oliveira, Richard Dahl, Judith A. Smith.

**Writing – original draft:** Mike Khan.

**Writing – review & editing:** Mike Khan, Jerome S. Harms, Sergio Costa Oliveira, Richard Dahl, Gary A. Splitter, Judith A. Smith.

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
