## [Decision Letter · Decision Letter 0]

20 Apr 2020

Dear Dr. Smith,

Thank you very much for submitting your manuscript "Brucella suppress innate immunity by down-regulating STING expression in macrophages" for consideration at PLOS Pathogens. We apologize for the long time in review--many of our editors and reviewers have been pulled into the COVID-19 response and we are encountering significant delays in review. As with all papers reviewed by the journal, your manuscript was reviewed by members of the editorial board and by several independent reviewers. In light of the reviews (below this email), we would like to invite the resubmission of a significantly-revised version that takes into account the reviewers' comments.

We cannot make any decision about publication until we have seen the revised manuscript and your response to the reviewers' comments. Your revised manuscript is also likely to be sent to reviewers for further evaluation.

Please prepare and submit your revised manuscript within 60 days if you are currently able to do experiments, or let us know if your university is closed and you will need longer to complete the experiments, and we will extend the time for you.

Sincerely,

Renée Tsolis

Section Editor

PLOS Pathogens

Kasturi Haldar

Editor-in-Chief

PLOS Pathogens

orcid.org/0000-0001-5065-158X

Michael Malim

Editor-in-Chief

PLOS Pathogens

orcid.org/0000-0002-7699-2064

Reviewer's Responses to Questions

**Part I - Summary**

Reviewer #1: Brucella suppress innate immunity by down-regulating STING expression in macrophages

Mike Khan and colleagues have investigated the effect of the microRNA, miR-24-2, on Tmem173 expression, the gene encoding the innate immune sensor STING, during Brucella infection. Decreased STING results in decreased type I IFNs and increased bacterial numbers in macrophages. In vivo, STING-/- mice had increased bacterial burdens in the spleen at 3 and 6 weeks post infection compared to wildtype infected mice. The authors show that during infection the expression of Tmem173 is reduced, and this correlates with the increased expression of miR24-2. Inhibition of miR24 using anti miRs and mirn23a locus deficient macrophages resulted in unaltered Tmem173 expression and subsequently reduced intracellular Brucella. Altogether, the data demonstrate that Brucella actively upregulate miR-24 expression to repress Tmem173 expression, resulting in increased bacterial numbers and chronic Brucella infection.

The manuscript is well written and the experiments well executed with appropriate controls. The major concern is the novelty of the data presented in this manuscript. It has been well established that miR-24 represses Tmem173 expression (PMID: 22387590 and 31735332). Furthermore, the role of STING in limiting Brucella infection has also been previously demonstrated. The innovative part of this manuscript is the active upregulation of miR-24 by Brucella which requires a functional T4SS. Adding more dating to this manuscript to investigate the mechanism of Brucella-induced upregulation of miR-24 (possibly mediated by an effector protein, or intracellular PAMPs (e.g. LPS) independent of MyD88 signaling) would greatly enhance this manuscript. Furthermore, all experiments (except Figure 1B) are performed with immortalized BMDMs and in vivo evidence (mouse infections) is lacking showing that Brucella upregulates miR-24, resulting in STING suppression and increased bacterial numbers. Mirn23a-/- mice should be available to the authors, since mirn23a-/- macrophages were used in this study. What is the effect of mirn23a deficiency on Brucella infections (chronic and/or acute) in vivo?

Minor comments:

Figure 3B. An intact T4SS is required for Tmem173 expression, but does deletion of the T4SS result in lower numbers of intracellular bacteria that could account for the partial rescue of Tmem173 expression? Possibly the amount of intracellular bacteria, or PAMPs, could lead to miR-24 upregulation?

The authors should rephrase sentence 162/163. Statistically there was no significant increase of miR-24 in Brucella infected MyD88-/- macrophages compared to uninfected MyD88-/- macrophages, suggesting MyD88 might be required in miR-24 expression (even though there was still a small increase in miR-24 expression in MyD88-/- macrophages).

Reviewer #2: This study by Khan and collaborators investigates the underlying mechanisms of STING down-modulation upon Brucella infection pinpointing to the role of a STING-targeting microRNA, miR-24-2 as a key regulator of STING expression induced by Brucella. The study is well-conducted and follows a hypothesis-driven rationale. Results are novel in the field and will serve to improve the knowledge of how this zoonotic pathogen important for human health deals with innate immunity to survive inside macrophages

Anyway, some concerns are raised

Reviewer #3: Khan et al present very strong and clear study showing Brucella suppresses STING expression via induction of a specific microRNA, which favors intracellular replication. The manuscript is very well written and for the most part data support the conclusions. Data are novel and of significance in the field.

**Part II – Major Issues: Key Experiments Required for Acceptance**

Reviewer #1: As outlined above (Part I), the authors should 1) investigate the mechanism of Brucella-induced expression of miR-24 and 2) provide evidence of the in vivo (mouse experiments) relevance of miR-24 expression during Brucella infections.

Reviewer #2: 1) Regarding the results show in Figure 1 respect to the role of STING in the control of Brucella infection in vivo and in vitro. Results seem to be mentioned as complementary of previous studies but to really know if the control of infection begins at early time points (as it seems to this reviewer, and is suggested by the authors in the discussion section, lines 273-274) or late in the chronic phase of infection; as the experiments performed in vivo in this Figure and stated in the title of Results. Thus, a kinetic should be performed in vitro (12, 24, 72 h) and in vivo (1, 3 and 6 wk) under the same experimental conditions to understand how early or late STING is controlling Brucella infection.

2) Scramble control of transfection with anti Mir-24 used to show specificity does not work properly. Levels of mir24 expression should be comparable to the ones achieved with infection in the absence of specific mir-24 inhibition, thus invalidatin the results and conclusions of Fig1 s and more important results in Figure 6. Please perform again the experiments (Fig1 S and Fig 6) using another scramble control that allows clear conclusions on the results obtained.

3) Although the use of v-raf/v105 myc immortalized murine bone marrow-derived macrophages is convenient and reliable, at least in when they need to be transfected. Some experiments would favor if primary BMDM are use. For instance, downregulation od STING expression do to Brucella species and the requirement of live bacteria and TSS4 system shoul be performed in WT BMDM to demonstrate the universality of the phenomenon.

Reviewer #3: I really only have one concern. In my opinion, with the experiments that were carried out authors cannot conclude the phenotypes are dependent on the T4SS. At 24h, the virB mutant is highly attenuated so there is 1) a significant difference in number of bacteria in cells infected between the wt and mutant and 2) the mutant is in a completely different intracellular compartment. Both of these could indirectly impact the results and give the false impression that the T4SS is necessary.

Authors could try to do these experiments at earlier time-points were trafficking and CFU counts are comparable between the WT and the virB mutant or simply remove these data, which are not essential for any of the conclusions presented.

**Part III – Minor Issues: Editorial and Data Presentation Modifications**

Reviewer #1: See above in Part I

Reviewer #2: 1) Title does not reflect the results obtained in the investigation. It should refer to the role of mir24 in STING down-modulation. Indicating the role of STING in innate immunity is an overstatement. Also the role of STING in innate immune responses has been already investigated and published by Costa Franco MM et al (J Immunol. 2018 ;200(2):607)

2) Figure 2 A: why STING expression was evaluated a number of very different times depending upon the species investigated? Melitensis 25 times, Abortus 7 and the other 2 species 5 times ….Results does not seem to be variable? Please explain.

3) Figure 2B: please show a more representative WB that the one depicted. Beta-acting seems to vary in the different lanes. Since the experiment was repeated several times as referred in the Figure legend a more representative photo must be shown. Why the picture has 3 lanes when the bar graph only has 2?

4) Figure 3 C: It is not clear what the asterisks in the HK bar stand for? In the texts is stated that HK is “comparable “to WT (line 120). Please clarify

5) Overall figure legends are no easy to follow. Modify them to make them fit with what is shown in the figures

6) Figure 4 A: Why if in the text (line 147) states that mir-24 levels vary between 50% and 8 times, the dispersion shown in the figure is so small (4 +/- 1 times). What kind of replicate were included in the data show?.

7) Discussion: Lines 304-305 are an overstatement to say that “Evades full activation of innate immunity”. It should say modifies IFN beta expression and apoptosis at the most

Reviewer #3: none

PLOS authors have the option to publish the peer review history of their article (what does this mean?). If published, this will include your full peer review and any attached files.

Reviewer #1: No

Reviewer #2: No

Reviewer #3: No
---

## [Editor Report · Decision Letter 1]

1 Oct 2020

Dear Dr. Smith,

We are pleased to inform you that your manuscript 'Brucella suppress STING expression via miR-24 to enhance infection' has been provisionally accepted for publication in PLOS Pathogens.

Best regards,

Renée M. Tsolis

Section Editor

PLOS Pathogens

Renée Tsolis

Section Editor

PLOS Pathogens

Kasturi Haldar

Editor-in-Chief

PLOS Pathogens

orcid.org/0000-0001-5065-158X

Michael Malim

Editor-in-Chief

PLOS Pathogens

orcid.org/0000-0002-7699-2064

Dear Judy, my sincere apologies for the long time in review. We had two different editors handling this manuscript, and it fell between the cracks--I'm very sorry. You've done a great job responding to the review and I think it has improved the paper!
---

## [Editor Report · Acceptance letter]

18 Oct 2020

Dear Dr. Smith,

We are delighted to inform you that your manuscript, "*Brucella* suppress STING expression via miR-24 to enhance infection," has been formally accepted for publication in PLOS Pathogens.

Best regards,

Kasturi Haldar

Editor-in-Chief

PLOS Pathogens

orcid.org/0000-0001-5065-158X

Michael Malim

Editor-in-Chief

PLOS Pathogens

orcid.org/0000-0002-7699-2064